Subject Areas:
materials science/physical chemistry

Keywords:
hyperbranched polymers, thermal degradation, thermogravimetric analysis, decomposition kinetics, degradation mechanism, rheology

Author for correspondence:
Norazilawati Muhamad Sarih
e-mail: nmsarih@um.edu.my

This article has been edited by the Royal Society of Chemistry, including the commissioning, peer review process and editorial aspects up to the point of acceptance.

# Rheological and thermal degradation properties of hyperbranched polyisoprene prepared by anionic polymerization

Shehu Habibu[1,2], Norazilawati Muhamad Sarih[1], Nor Asrina Sairi[1] and Muzafar Zulkifli[3,4]

[1]Department of Chemistry, Faculty of Science, University of Malaya, 50603 Kuala Lumpur, Malaysia
[2]Department of Chemistry, Faculty of Science, Federal University Dutse, PMB 7651, Jigawa, Nigeria
[3]Section of Polymer Engineering Technology, and [4]Green Chemistry and Sustainable Engineering Technology Cluster, Universiti Kuala Lumpur, Branch Campus Malaysian Institute of Chemical and Bioengineering Technology (UniKL MICET), Lot 1988, Taboh Naning, 78000 Alor Gajah, Malacca, Malaysia

SH, 0000-0002-3746-6643

Hyperbranched polyisoprene was prepared by anionic copolymerization under high vacuum condition. Size exclusion chromatography was used to characterize the molecular weight and branching nature of these polymers. The characterization by differential scanning calorimetry and melt rheology indicated lower $T_g$ and complex viscosity in the branched polymers as compared with the linear polymer. Degradation kinetics of these polymers was explored using thermogravimetric analysis via non-isothermal techniques. The polymers were heated under nitrogen from ambient temperature to 600°C using heating rates from 2 to 15°C min$^{-1}$. Three kinetics methods namely Friedman, Flynn–Wall–Ozawa and Kissinger–Akahira–Sunose were used to evaluate the dependence of activation energy ($E_a$) on conversion ($\alpha$). The hyperbranched polyisoprene decomposed via multistep mechanism as manifested by the nonlinear relationship between $\alpha$ and $E_a$ while the linear polymer exhibited a decline in $E_a$ at higher conversions. The average $E_a$ values range from 258 to 330 kJ mol$^{-1}$ for the linear, and from 260 to 320 kJ mol$^{-1}$ for the branched polymers. The thermal degradation of the polymers studied involved one-dimensional diffusion mechanism as determined by Coats–Redfern method. This study may help in understanding the effect of branching on the rheological and decomposition kinetics of polyisoprene.

# 1. Introduction

Hyperbranched polymers (HBPs) have received attention for many years due to the favourable properties they possess over their linear counterparts as well as ease of synthesis when compared with dendrimers. Strathclyde methodology is one of the ways in which the hyperbranched polymer synthesis could be achieved. It involves the copolymerization of vinyl monomers with the help of chain transfer agent that can prevent the formation of gel [1,2]. Understanding the effect of molecular architecture on the properties of polymers is of interest from both theoretical and practical perspectives [3]. Hyperbranched and highly branched polymers represent a class of materials which differ significantly in their chemical compositions as well as physical properties when compared with the linear polymers. HBPs are widely used in various fields such as coatings, nanotechnology, supramolecular chemistry, drug delivery systems, and as thermal and viscosity modifiers due to their lower cost as well as ease of synthesis compared with the dendrimers [4,5]. Recently, researchers have focused on the characterization of HBPs, particularly their structure–property relations, such as morphology, branching density, solution properties, crystallinity and glass transition temperature [4]. Many reports have been published on the chemical and physical changes that occurred when polymers were exposed to heat over a wide range of temperature [6–9]. Rheology of branched polymers is a special topic of interest to both industry and academia due to the correlation between rheological properties and the degree of branching of polymers [10]. The rheology of polymers of various architectures ranging from linear, star and dendrimers has been investigated by various groups [11–17]. The degree of branching of polymers strongly influenced their rheological properties and processing ability [14]. To well understand the correlation between branching and rheology of HBPs is therefore strongly motivated.

Thermal analysis of polymers is very crucial in understanding the molecular architecture, degradation and mechanisms, and in determining their application under various environmental conditions such as at elevated temperature. Extensive research on thermal degradation mechanism of the presently available macromolecules is available in the literature [18–22]. However, these studies have focused on thermal properties of commercial homopolymers or blends of polymers. Moreover, there are few reports on this aspect related to hyperbranched and highly branched polymers. Othman *et al.* [23] gave some insight into the effect of terminal groups on the thermal decomposition of some flame-retardant hyperbranched polyimides [23]. Chen and his co-workers [24] studied the thermal degradation properties of the hyperbranched exopolysaccharide using the isoconversional method of Flynn–Wall–Ozawa (FWO) followed by the Coats–Redfern method to model the thermal degradation of these HBPs. Röchow *et al.* [25] explored the thermal decomposition behaviour of methyl methacrylate (MMA) copolymer with a phosphate-containing comonomer, ethyl-2-[4-(dihydroxyphosphoryl)-2-oxabutyl]acrylate (EDHPOBA) and observed a more complex decomposition mechanism when compared with the homopolymers of MMA. Methods used for evaluating the kinetic parameters of various physico-chemical processes from thermal analysis data can be classified into model-fitting or model-free techniques [26].

Principally, a fixed mechanism is assumed in the model-fitting approach with constant activation energy throughout the reaction. This technique consists of the fitting of kinetic data to some models to compute the kinetic parameters such as activation energy and the order of reaction. However, it is difficult to model complex reactions with reasonable accuracy. This is a limitation associated with the model-fitting approach. Model-free kinetics involves an isoconversional analysis on kinetic data obtained at a minimum of three heating rates, and activation energy is varied with temperature [27–29]. Thus, in model-free approaches, more than one mechanism is allowed during the course of the reaction. The main disadvantage of the model-free method is the absence of a reaction model which is required for a complete kinetic description to be achieved. Regardless of which of the two approaches, model-free or model fitting, is employed, it is unanimously assumed that the reaction rate has an Arrhenius-type dependence on temperature [30]. The primary objective of the Arrhenius-type of kinetic expressions is to determine the kinetic triplets experimentally. In the isoconversional methods, the complexity of a reaction is revealed via a functional interdependence between the activation energy and the extent of reaction ($\alpha$). This approach allows for a good prediction of kinetic information. When the activation energy of a process varies significantly with the conversion, it indicates that the process is complicated from a kinetic point of view.

In this study, HBPs were prepared via anionic polymerization based on the Strathclyde approach. The rheological properties of HBPs were investigated under oscillatory shear experiments, and a comparison was made with a linear polymer sample. The information obtained from the TEM analysis could also assist in establishing and comparing the morphologies of the branched and the linear polymers.

Thermogravimetric analysis (TGA) was used to study the thermal degradation kinetics of hyperbranched polyisoprene. The experiments were conducted under nitrogen at 2, 5, 10 and 15°C min$^{-1}$. The objective was to obtain the kinetic parameters for the thermal degradation using four isoconversional methods, namely the Kissinger method, the Friedman method, Flynn–Wall–Ozawa (FWO) and Kissinger–Akahira–Sunose (KAS) methods. The relationship between the extent of conversion ($\alpha$) and the activation energy ($E_a$) has also been discussed, and finally, the Coats–Redfern method was employed to determine the mechanism of thermal degradation of these polymers.

# 2. Experimental procedure

## 2.1. Materials

Isoprene (99%), and the initiators (1.4 M $s$-butyl lithium in cyclohexane, and 2.0 M $n$-butyllithium in cyclohexane, as well as the 99.5% N,N,N,N-tetramethylenediamine (TMEDA)) were obtained from Sigma-Aldrich; 99.9% dried methanol, and 98% divinylbenzene (DVB) were purchased from Merck (Germany); 99% 2,6-di-tert-butyl-4-methylphenol (BHT) was obtained from Fischer Scientific. The solvents (99.9% benzene and HPLC grade toluene) and the monomers were purified through over CaH$_2$ (Aldrich) via several freeze–pump–thaw cycles on a vacuum line. All the syntheses were carried out using a special reactor ('Christmas tree' reaction vessel), which was purified according to the similar procedure reported by Habibu et al. [1] and Hutchings et al. [31].

## 2.2. Synthesis

The polymers were prepared via anionic polymerization technique under high vacuum conditions similar to the one previously reported [31]. About 100 ml of the solvent was collected into the reactor by vacuum distillation, and the monomer was added with the help of an attachable flask. Before transferring the monomer to the reactor, 0.1 ml of $n$-butyllithium to it. Later, TMEDA, which serves to promote chain transfer, was added onto the mixture of the solvent and the monomer, before initiation with $s$-butyllithium. About 5 min after initiation, DVB was injected, and the reaction proceeded over night at constant temperature of 50°C. The reaction was terminated with dried methanol, and the polymer was precipitated in excess methanol. $^1$H NMR (400 MHz; CDCl$_3$; Me$_4$Si) 7.13–7.18 (3H, m, Ar), $\delta$H 4.94–5.12 (3H, m, 1,4 motifs, olefinic), 4.77–4.94 (3H, m, 1,2), 4.67–4.76 (5H, m, 3,4), 4.64 (7H, br. s., 3,4), 2.26 (1H, s, 3,4 motifs, aliphatic), 1.79–2.06 (14H, m, 1,4, aliphatic).

## 2.3. Characterization

The molecular weight and the dispersity index (Đ) of the polymers were obtained by gel permeation chromatography (GPC, Viscotek 302, Malvern) system equipped with light scattering, viscosity and refractive index detectors. The elution of the columns (300 mm PLgel, 5 mm mixed C) was achieved using tetrahydrofuran (THF) at 35°C, and 1 ml min$^{-1}$ as the flow rate. Polystyrene standards were employed to calibrate the detectors and the refractive index of 0.087 and 0.127 ml g$^{-1}$ were used for the branched and the linear poly(isoprene), respectively. For the analysis, 5.0 mg ml$^{-1}$ of the samples were prepared in THF. Table 2 presents details on the molecular weight characteristics. $^1$H NMR analysis was performed in CDCl$_3$ on a DELTA2 (JEOL) 400 MHz spectrometer. Reference for the chemical shift was made to the traces of CHCl$_3$ (7.26 ppm) present in the CDCl$_3$.

## 2.4. Transmission electron microscopy

The morphology and structural features of the branched polymers were analysed using high-resolution transmission electron microscopy (HRTEM). HRTEM is a powerful technique used in polymer science to analyse the shape and structure of a polymeric material. HRTEM was used to investigate the effect of the branching on the morphology of the branched polymers and to compare them with linear polymers. Images were obtained by FEI TECNAI G2 F20 X–TWIN transmission electron microscope (TEM) operated at 200 kV. The analysis was performed at the Malaysian Institute of Microelectronic Systems (MIMOS). The samples were prepared and analysed according to a similar procedure to that reported by Hutchings and co-workers [32].

## 2.5. Rheological measurements

The rheological analysis was achieved on an Anton Paar MCR301 rheometer using a parallel plate geometry (25 mm) with a gap of 1 mm and a convection temperature device (CTD). Amplitude sweeps were initially conducted to determine the linear viscoelastic regime and the frequency sweep measurements were carried out from 0.1 rad s$^{-1}$ to 100 rad$^{-1}$ in the linear viscoelastic regime with a strain of 1% at 70°C. The storage and loss moduli, as well as complex viscosity, were measured.

## 2.6. Differential scanning calorimetry

The glass transition temperature ($T_g$) of the polymers was measured on a DSC 822e, Mettler Toledo calorimeter. About 4–8 mg of each sample was placed into an aluminium pan and the temperature was raised to 200°C at 10°C min$^{-1}$ under nitrogen at a flow rate of 20 ml min$^{-1}$. The polymer was then cooled down to −40°C, and second heating was made from −40°C to 200°C. Data from the second heating were used for the analysis. The data were processed using thermal analysis software (STARe).

## 2.7. Thermogravimetric analysis

Simultaneous thermal analyser, STA 6000 (Perkin Elmer) was used for TGA. About 10 mg of the polymer samples were put in a ceramic crucible, and the temperature was raised from 30°C to 600°C at 20 ml min$^{-1}$ nitrogen flow rate and at the rates of 2, 5, 10 and 15°C min$^{-1}$.

## 2.8. Theoretical background

The use of non-isothermal thermogravimetric methods has a great capacity to unravel the mechanisms of chemical and physical processes that occur during the degradation of polymers. The kinetic methods used in the thermal analysis of single-step reactions have been established over the years. For systems that involve multi-step reactions, the use of inappropriate kinetic method can lead to misleading results. Nevertheless, it has been revealed that the use of isoconversional methods can provide meaningful values of activation energy in a wide-ranging environment. The Isoconversional techniques allow for a model-free estimation of the activation energy. Furthermore, this method is based on the isoconversional principle which states that at a given extent of conversion, the decomposition rate solely depends on the current sample temperature. It has been concluded that model-free (isoconversional) methods are powerful and most reliable tools to estimate the activation energy of thermally stimulated processes [33–35]. In this research, three isoconversional methods, the Friedman, FWO and KAS methods, were employed to evaluate the dependence of activation energy ($E_a$) on conversion ($\alpha$) for these polymers.

The rate of thermal decomposition for the solid-state reaction d$\alpha$/d$t$ follows the equation

$$\frac{\mathrm{d}\alpha}{\mathrm{d}t} = k(T)f(\alpha), \tag{2.1}$$

where $k(T)$ is the temperature-dependent rate constant, $f(\alpha)$ is the kinetic model which depends on the particular decomposition mechanism, and $\alpha$ is the degree of conversion represented by the following equation:

$$\alpha = \frac{w_o - w_t}{w_o - w_f}, \tag{2.2}$$

where $w_0$ is the initial weight of the polymer sample, $w_f$ is the final weight, and $w_t$ is the weight at a given time of the experiment.

The temperature-dependent rate constant was assumed to follow the Arrhenius equation

$$k(T) = A\mathrm{e}^{-E_a/RT}, \tag{2.3}$$

where $E_a$ is the apparent activation energy (kJ mol$^{-1}$), $A$ is the pre-exponential factor (min$^{-1}$) and $R$ is the gas constant (8.3142 J mol$^{-1}$ K$^{-1}$). Combining equations (2.1) and (2.3) gives rise to equation (2.4) which is

the fundamental equation used to calculate the kinetic parameters based on the TGA.

$$\frac{d\alpha}{dt} = f(\alpha)Ae^{-E_a/RT}.$$

(2.4)

The most frequently used model for $f(\alpha)$ to analyse thermogravimetric data is given by

$$f(\alpha) = (1 - \alpha)^n,$$

(2.5)

in which $n$ represents the reaction order. Substituting equation (2.5) into (2.4) gives

$$\frac{d\alpha}{dt} = (1 - \alpha)^n Ae^{-E_a/RT}.$$

(2.6)

For a non-isothermal thermogravimetric experiment at a constant heating rate $\beta = d\alpha/dt$, the above equation can be written as

$$\frac{d\alpha}{dT} = \frac{A}{\beta}(1 - \alpha)^n e^{-E_a/RT}.$$

(2.7)

The above equation is a differential form of the non-isothermal rate law. In the present study, non-isothermal methods were used to calculate the kinetic parameters.

## 2.9. Friedman method

The Friedman method is the most commonly used differential isoconversional process [28,36]. The following equation usually describes the temperature-dependent rate constant ($k$):

$$kf(\alpha) = \beta\left(\frac{d\alpha}{dT}\right) = Af(\alpha)e^{-Ea/RT}.$$

(2.8)

Taking the logarithm of both sides of equation (2.8) followed by simple rearrangement gives Friedman's equation:

$$\ln\left(\beta\frac{d(\alpha)}{dT}\right) = \ln\left(\frac{d(\alpha)}{dt}\right) = \ln[Af(\alpha)] - \frac{E_a}{RT}.$$

(2.9)

The apparent activation energy ($E_a$) is obtained from the slope ($-E_a/R$) of a plot of $\ln(d\alpha/dt)$ against reciprocal temperature ($1000/T$) at constant conversion for a given set of heating rates.

## 2.10. Flynn–Wall–Ozawa method

Flynn, Wall and Ozawa proposed the FWO method [37–39]. The main advantage of the FWO method is that apart from the Arrhenius temperature-dependence equation, no assumptions regarding the form of the kinetic model equations are required [28]. The FWO method is an integral model-free method which involves the measurement of temperature that corresponds to a fixed value of conversion at different heating rates. Integration of equation (2.10) and applying the Doyle approximation give rise to equation (2.12) in which the plot of $\ln \beta$ against $1000/T$ provides a straight line with slope equal to $-1.052E_a/R$ [28,40].

$$\ln \beta = \ln\left(\frac{AE}{Rg(\alpha)}\right) = 5.331 - 1.052\left(\frac{E_\alpha}{RT}\right),$$

(2.10)

where the kinetic model is $g(\alpha)$ (i.e. $kt = g(\alpha)$). Therefore, when g($\alpha$) is known for a constant $\alpha$, plots of $\ln \beta$ against the $1000/T$ taken at different $\beta$ should give a straight line. The activation energy and the pre-exponential factor values are obtained from the slope and the intercept, respectively.

## 2.11. Kissinger–Akahira–Sunose method

KAS method is an isoconversional method in which for a given conversion ($\alpha$), the temperature ($T$) and the heating rate are related by the following equation [41,42]:

$$\ln\left(\frac{\beta}{T^2}\right) = \ln\left(\frac{AR}{E_a g(\alpha)}\right) - \left(\frac{E_a}{RT}\right).$$

(2.11)

**Table 1.** E expressions for $f(\alpha)$ for the most commonly used mechanisms of solid-state processes.

| Model | $f(\alpha)$ | mechanism |
|---|---|---|
| $A_2$ | $[-\ln(1-\alpha)]^{1/2}$ | nucleation and growth (Avrami equation 1) |
| $A_3$ | $[-\ln(1-\alpha)]^{1/3}$ | nucleation and growth (Avrami equation 2) |
| $A_4$ | $[-\ln(1-\alpha)]^{1/4}$ | nucleation and growth (Avrami equation 3) |
| $R_2$ | $[1-(1-\alpha)^{1/2}]$ | phase boundary-controlled reaction (contracting area) |
| $R_3$ | $[1-(1-\alpha)^{1/3}]$ | phase boundary-controlled reaction (contracting volume) |
| $D_1$ | $\alpha^2$ | one-dimensional diffusion |
| $D_2$ | $(1-\alpha)\ln(1-\alpha)+\alpha$ | two-dimensional diffusion |
| $D_3$ | $[1-(1-\alpha)^{1/3}]^2$ | three-dimensional diffusion (Jander equation) |
| $D_4$ | $(1-(2/3)\alpha)-(1-\alpha)^{2/3}$ | three-dimensional diffusion (Gingstling–Brounshtein equation) |
| $F_1$ | $-\ln(1-\alpha)$ | random nucleation with one nucleus on the individual particle |
| $F_2$ | $1/(1-\alpha)-1$ | random nucleation with two nuclei on the individual particle |
| $F_3$ | $1/(1-\alpha)^2-1$ | random nucleation with three nuclei on the individual particle |

In the KAS method, for each value of $\alpha$, a straight line plot of $\ln(\beta/T^2)$ against $1000/T$ is obtained from the non-isothermal data acquired at various $\beta$ [43].

## 2.12. Coats–Redfern method

This is an integral method that involves the mechanism for thermal degradation. The activation energy is calculated based on the $f(\alpha)$ functions according to equation (2.12) derived by applying an asymptotic approximation [44–46].

$$\ln\frac{g(\alpha)}{T^2} = \ln\frac{(AR)}{\beta E} - \frac{E}{RT}. \tag{2.12}$$

The Coats–Redfern method assumes that the activation energy does not depend on the degree of conversion. The activation energy for each model is acquired using the slope obtained from the plot of $\ln(g(\alpha)/T^2)$ against $1000/T$. Achievement of high correlation coefficient ($R^2$), as well as agreement between the activation energy obtained by this method compared with those obtained by previous methods, allows for the selection of the kinetic model [23,46,47]. The $f(\alpha)$ expressions for various mechanisms are listed in table 1 and the same values of $\alpha$ were used as in the isoconversional methods.

# 3. Results and discussion

The molecular weight and dispersity index for the linear and the branched polymers are presented table 2. The branched polymers and the linear (LP-02) possess higher dispersity index due to addition of TMEDA, which is absent in the case of the linear polymer (LPI-01) [48,49]. The [1]H NMR spectrum is presented in figure 1. The characteristic peaks of the alkene protons of the polyisoprene exist in the range 4.5–5.5 ppm.

The peaks at 5–5.2 ppm corresponds to 1,4-microstructure [$(CH_3)C=CH$], while those at 4.6–4.8 ppm are attributed to the 3,4-microstructure [$(CH_3)C=CH_2$]. Difference in the microstructure between the linear and the branched polymers exist due to the addition of TMEDA. The linear polymer consists mainly of 1,4-microstructure, whereas the highly branched samples are pre-dominantly 3,4- (figure 1). Peak $a$ is due to 1,4- microstructure, while peaks $b$ and $c$ are due to 3,4- and 1,2- microstructure, respectively. Peak $c$ is absent in LPI-01 with TMEDA/Li = 0, and there is appearance of very small $b$, whereas in HBPI-01 with TMEDA/Li = 0.5, there is appearance of very small peak $c$, which is more evident in HBPI-02 with TMEDA/Li = 1.5. Similarly, peak $b$ is more evident in HBPI-01 due to more TMEDA content.

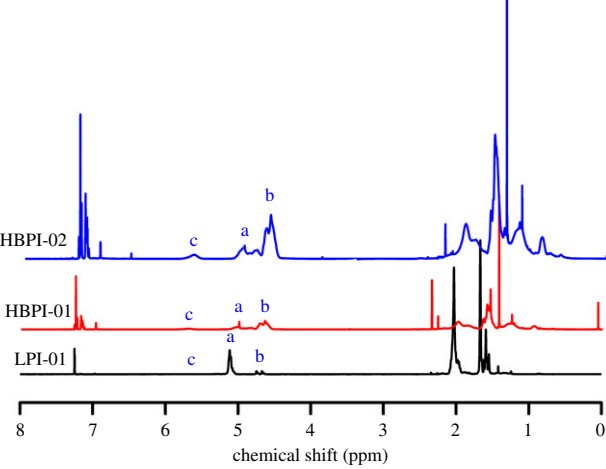

**Figure 1.** [1]H NMR spectra of the linear and hyperbranched polyisoprene.

**Table 2.** Molecular characteristics of the linear and hyperbranched polyisoprene.

| sample | TMEDA : Li | DVB : Li | $M_n$(g/mol) | Đ | $[\eta]_{br}$(dlg$^{-1}$)[a] | $[\eta]_{lin}$(dlg$^{-1}$)[b] | $G'$[c] | $T_g$ (°C) |
|---|---|---|---|---|---|---|---|---|
| LPI-01 | 0 | 0 | 19 100 | 1.04 | — | — | — | −15.6 |
| LPI-02 | 1.0 | 0 | 9300 | 2.25 | — | — | — | −13.6 |
| HBPI-01 | 0.5 | 1.0 | 17 900 | 2.04 | 0.24 | 0.40 | 0.58 | −19.5 |
| HBPI-02 | 1.5 | 3.0 | 16 900 | 6.41 | 0.32 | 0.89 | 0.37 | −23.3 |
| HBPI-03 | 0.5 | 2.0 | 82 400 | 1.87 | 0.48 | 1.15 | 0.42 | −12.7 |
| HBPI-04 | 1.0 | 3.0 | 141 500 | 2.19 | 0.2871 | 1.926 | 0.15 | −9.52 |

[a]Measured by size exclusion chromatography in THF at 35°C.

[b]Calculated using Mark–Houwink–Sakurada equation: $[\eta]_{lin} = K\, M_w^{\alpha}$, $K = 0.000177$ dlg$^{-1}$, $\alpha = 0.735$ dlg$^{-1}$.

[c]$G' = [\eta]_{hyper}/[\eta]_{lin}$, Đ = dispersity index ($M_w/M_n$).

## 3.1. Transmission electron microscopy

TEM has been widely used for many years to obtain information on the morphology of synthetic macromolecules. To assess the impact of branching on the morphology of the polymers as well as to have a comparison between the linear and branched polymer samples, we carried out TEM analysis on the linear polymer the (LPI-01) and branched polymer (HBPI-02). The investigation was carried out using HRTEM, and the samples were prepared for imaging by cryo-ultramicrotomy. Typical data are presented in figure 2*a* and *b* for the LPI-01 and HBPI-02, respectively. From figure 2, it is clear that the structure affects the morphology of these polymers. The increase in branching resulted in a less ordered morphology. It can be seen that the linear polymer shows a high level of long-range order as expected [50]. However, the morphology of the branched polymers is dramatically different. Differences in degree of branching between these polymer samples have a role to play in the morphologies of these polymers. High degree of branching obstructs any long-range ordering [51]. A similar decrease in the long-range order was observed upon introduction of branched points into 'barbed wires' built from polystyrene arms and polyisoprene backbones. Beyer *et al.* [52] reported a similar lack of well-ordered morphology in multigraft copolymers with a high degree of branching [51–56]. The disordered morphology could also occur due to the high dispersity of polymers [57,58]. Also, the results reported in the present study agree with those obtained previously by Hutchings *et al.* [59] in which they observed similar morphological changes upon conversion of macromonomers into hyperblocks. It was established that irrespective of the molecular weights or structures of the linear precursors, the resulting branched polymers lack the long-range lattice order associated with the branching which impedes the formation of a well-defined long-range lattice order [59].

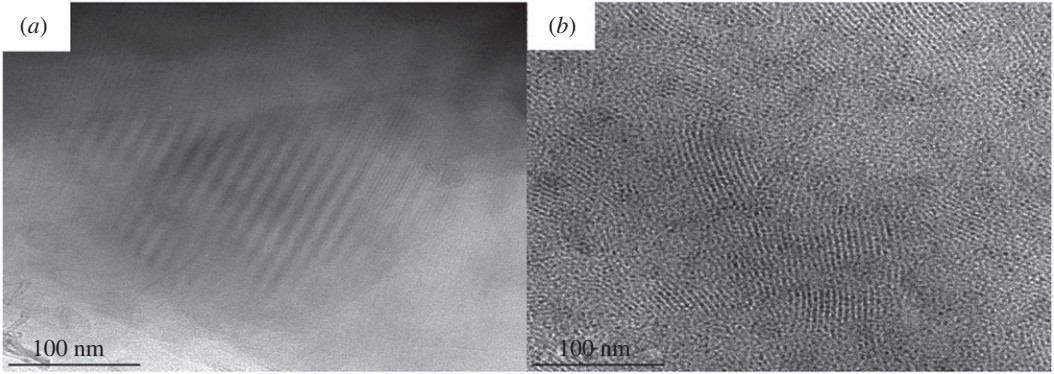

**Figure 2.** TEM images of (*a*) linear (LPI-01) and (*b*) branched (HBPI-02) polymers.

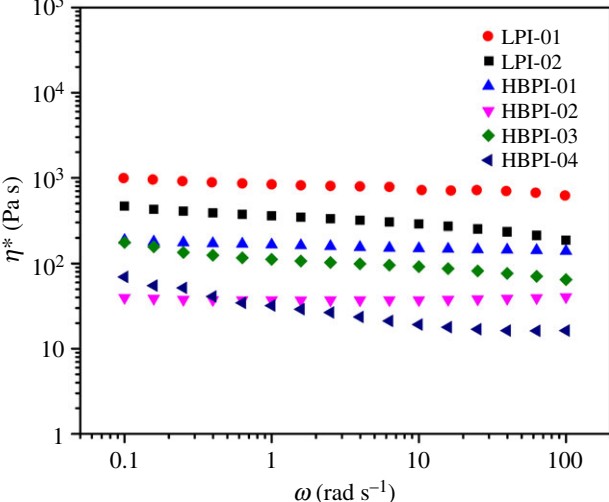

**Figure 3.** Complex viscosity versus angular frequency of the linear (LPI-01, LPI-02) and branched (HBPI-01, HBPI-02, HBPI-03, HBPI-04) polymers.

## 3.2. Melt rheological properties

The melt rheological properties of the branched polymers were evaluated using an MCR rheometer in oscillatory shear experiments at 70°C, and there was a strong correlation between the rheological properties of these polymers and their structures. The structural features of a polymer such as branching, molecular weight and the dispersity index (Đ), usually affects its rheological properties. Figure 3 shows the relationship between complex viscosity and angular frequency for the linear (LPI-01, LPI-02), and the branched (HBPI-01, HBPI-02 HBPI-03 and HBPI-04) polymer samples. At any given angular frequency, the linear polymer exhibited higher complex viscosity than the branched polymers [49]. Moreover, the branched polymers showed evidence of shear thinning behaviour compared with the linear polymer sample, and this indicates that the branched polymers are less entangled than their linear counterpart. Also, the viscosity of the HBPI-02 and HBPI-04 is lower than that of the HBPI-01 and HBPI-03 due to higher branching as well as dispersity index. The complex viscosity of HBPI-03 is higher than that of HBPI-02 due to higher molecular weight as well as lower molecular weight distribution. Also, the branched polymers almost exhibited constant complex viscosity, and this is also an indication of the absence or low chain entanglement. The loss ($G''$) and the storage ($G'$) moduli, respectively, measure the viscosity and elasticity of materials [16,49]. Figure 4 presents the loss modulus and the storage modulus as a function of the angular frequency, and it could be observed that the *y*-axis for the LPI-01 and LPI-02 extends to about 100 000 Pa, whereas for the HBPI-01 HBPI-02, HBPI-03 and HBPI-04, the *y*-axis extends only to up to 10 000 Pa. Moreover, the storage and loss moduli vary approximately in a linear fashion in the case of the branched polymers.

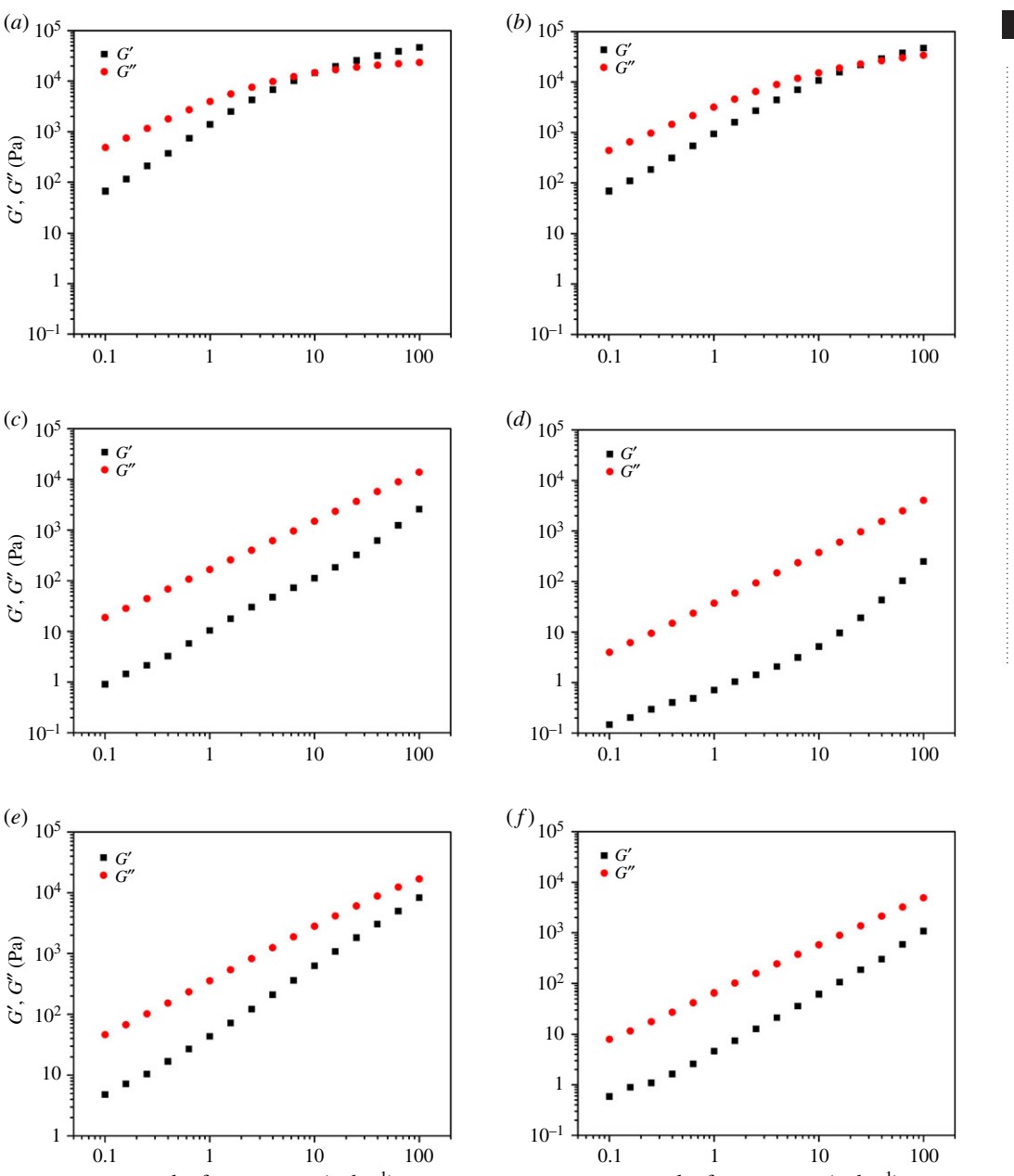

**Figure 4.** Storage modulus ($G'$), loss modulus ($G''$) versus angular frequency for the linear (a) LPI-01, (b) LPI-02 and branched (c) HBPI-01, (d) HBPI-02 (e) HBPI-03, (f) HBPI-04 polymers.

## 3.3. Differential scanning calorimetry

Segmental mobility of polymer chains is believed to influence the glass transition of polymers. Branched polymers possess a lower glass transition temperature ($T_g$) due to the presence of more free volume as a result of numerous end groups [1,49]. The differential scanning calorimetry (DSC) data presented in table 2 show that the linear samples (LPI-01 and LPI-02) exhibited higher values of $T_g$ (−15.6°C) than the branched polymers, (HBPI-01, −19.5) and (HBPI-02, −23.3°C), due to the higher chain mobility of the branched polymers over the linear polymer [60]. A similar result was reported in the literature [61]. Also, HBPI-02 exhibited a lower $T_g$ value due to the presence of soft segments which offers more mobility to the polymer chains due to high branching density [62]. Therefore, lower glass transition temperature occurs due to the higher degree of branching in HBPI-02 as compared with HBPI-01, although they possess roughly the same average molecular weight ($M_n$). However, higher $T_g$ values were observed for HBPI-03 due to high molecular weight and low dispersity compared with other branched and linear samples.

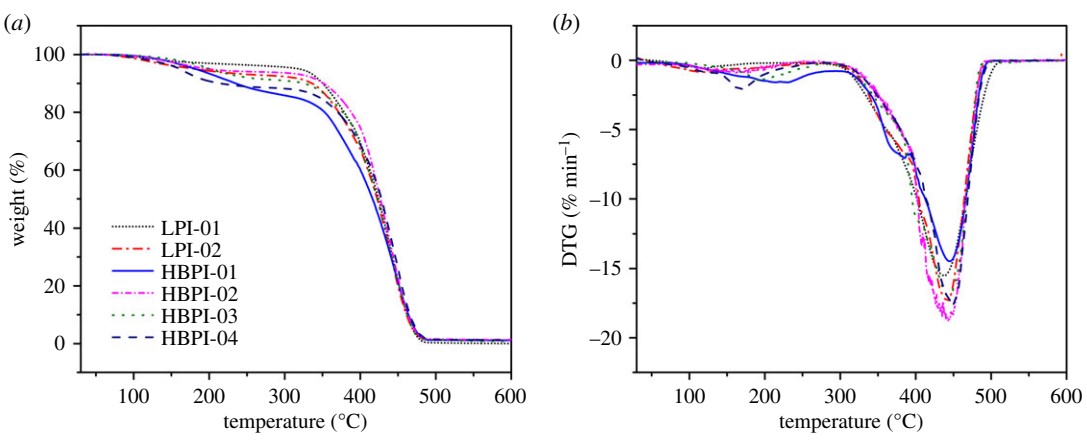

**Figure 5.** TGA (*a*) and DTG (*b*) curves at 15°C min$^{-1}$ for LPI-01, HBPI-01, HBPI-02 and HBPI-03.

**Table 3.** Thermal decomposition characteristics for the linear and hyperbranched polyisoprene.

| Title 1 | $T_{20\%}$ (°C) | $T_{50\%}$ (°C) | $T_{max}$ (°C) | % residue at 580°C |
|---|---|---|---|---|
| LPI-01 | 379 | 424 | 439 | 0.2 |
| LPI-02 | 371 | 423 | 441 | 1.1 |
| HBPI-01 | 388 | 427 | 444 | 1.3 |
| HBPI-02 | 377 | 423 | 445 | 0.8 |
| HBPI-03 | 353 | 417 | 446 | 1.1 |
| HBPI-04 | 371 | 427 | 448 | 1.2 |

## 3.4. Thermal degradation characteristics

TGA plots express the relationship between the percentage mass of the experimental samples and temperature. The mass loss–temperature curve allows for the direct evaluation of the temperatures for the initial and final degradation as well as the residual mass of the sample. Figure 5*a* shows a typical TGA curve obtained from the analysis of the linear and branched polymers heated at 15°C min$^{-1}$. It showed that the branching, as well as molecular weight distribution, affect the thermal stability of these polymers. The thermal stability of the polymers decreases with branching. Moreover, an increase in the dispersity lowers the thermal stability of polymers [45]. The low molecular weight components of more disperse samples could be the possible reason for the lower thermal stability of HBPI-02 compared with HBPI-01. It could be seen from figure 5 that after about 580°C, over 98% conversion was achieved and there was less than 2% residue left in all the three samples. Moreover, at higher heating rates there is a delay in the degradation process and the curves shift to higher temperatures.

The temperature of maximum decomposition ($T_{max}$) appears at 439°C for LPI-01 ($M_n = 19\,118$ g mol$^{-1}$, Đ = 1.04), for LPI-02 ($M_n = 9300$ g mol$^{-1}$, Đ = 2.25) it appeared at 441°C, while for HBPI-01 (DVB/Li = 1.0, $M_n = 17\,900$ g mol$^{-1}$ Đ = 2.04) appeared at about 444°C and that of HBPI-02 (DVB/Li = 3.0, Mn = 16\,900 g mol$^{-1}$ Đ = 6.41) was about 445°C. The $T_{max}$ for HBPI-03 (DVB/Li = 2.0, $M_n = 82\,400$ g mol$^{-1}$ Đ = 1.87) and HBPI-04 (DVB/Li = 3.0, $M_n = 141\,500$ g mol$^{-1}$ Đ = 2.19), were, respectively, 446°C and 448°C. Details of values are given in table 3. The more branched polymer sample (HBPI-04) possessed the higher temperature of maximum decomposition compared with the LPI-01, LPI-02 HBPI-01, HBPI-02 and HBPI-03 despite having the high dispersity index and this may be related to the higher DVB content. The highest $T_{max}$ observed in HBPI-04 may be related to its high molecular weight.

## 3.5. Kinetics of thermal degradation

The use of non-isothermal thermogravimetric methods has a high capacity to unravel the mechanisms of chemical and physical processes that occur in the course of the degradation of polymers. It has been revealed that the use of isoconversional approaches can provide meaningful $E_a$ values in different experimental conditions. Model-free evaluation of the activation energy is achieved via the

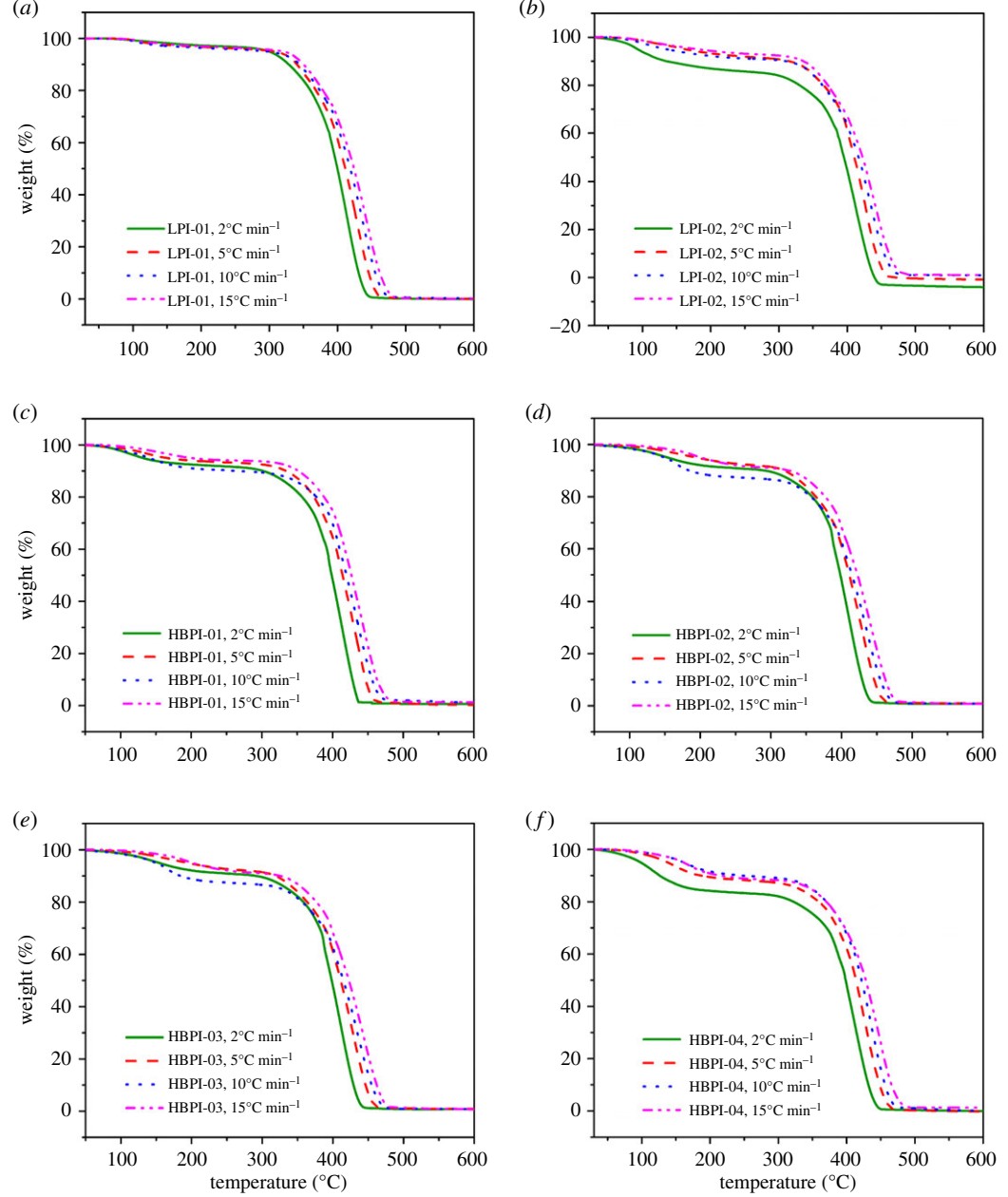

**Figure 6.** TGA plots at different heating rates for (*a*) LPI-01, (*b*) LPI-02, (*c*) HBPI-01, (*d*) HBPI-02, (*e*) HBPI-03 and (*f*) HBPI-04.

**Table 4.** Average activation energy ($E_a$) values obtained via the three kinetic methods.

| methods | average activation energy (kJ mol$^{-1}$) | | |
| --- | --- | --- | --- |
| | Friedman | FWO | KAS |
| LPI-01 | 330 | 319 | 324 |
| LPI-02 | 263 | 258 | 278 |
| HBPI-01 | 279 | 270 | 273 |
| HBPI-02 | 268 | 260 | 262 |
| HBPI-03 | 320 | 309 | 315 |
| HBPI-04 | 244 | 237 | 238 |

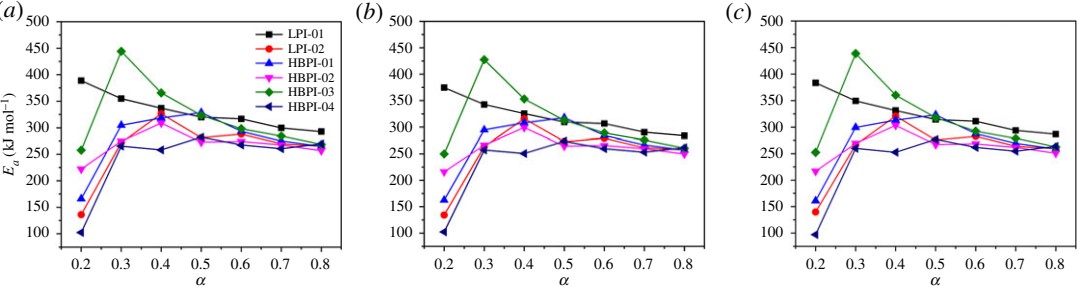

**Figure 7.** Activation energy against the extent of conversion obtained by (a) Friedman, (b) FWO and (c) KAS methods.

isoconversional techniques. The isoconversional methods are powerful and most reliable tools for estimating the activation energies of thermally stimulated reactions [33–35].

Three isoconversional methods which include the Friedman, FWO and KAS were selected to analyse the experimental results obtained from the thermal degradation kinetics due to their ability to estimate the kinetic parameters with good accuracy and simplicity [28,36,42,63]. Moreover, using these methods, the dependence of activation energy ($E_a$) on conversion ($\alpha$) for these polymers was evaluated. Figure 6a–f shows the typical TGA curves at different $\beta$ for LPI-01, LPI-02, HBPI-01, HBPI-02, HBPI-03, and HBPI-04, respectively.

The Friedman method which is a derivative method and the most commonly used differential isoconversional process [28,36,64] was first employed to obtain the kinetic information of the polymers. This technique correlates the conversion rate ($\alpha$) with various heating rates at a specified conversion. Equation (2.9) has been employed to obtain the $E_a$ values by plotting the $\ln[\beta(d\alpha/dt)]$ versus $1/T$ for the conversion range ($\alpha = 0.2$–08) and the results are summarized in table 4. The average activation energies for the branched polymers (HBPI-01, HBPI-02, HBPI-03 and HBPI-04) were lower compared with the one obtained for the linear polymer (LPI-01). Also, the average activation energy for HBPI-02 was lower compared with the HBPI-01 with less degree of branching, but HBPI-03, although possessing a higher degree of branching, exhibited higher $E_a$ due to high molecular weight. HBPI-04 with the highest degree of branching exhibited the lower average activation energy values.

The second method was the FWO which is an integral and model-free method. Equation (2.10) was employed, and the $E_a$ was obtained from the plots of $\ln\beta$ against $1/T$ for a fixed conversion ($\alpha$). The $E_a$ was obtained from the slope which equals to $-1.052 E_a/R$ [28,40,65]. Similar values of $\alpha$ were used as in the case of Friedman method. The average $E_a$ values for the LPI-01, HBPI-01 and HBPI-02 are given in table 4. The average values attained by the FWO method for LPI-01, LPI-02, HBPI-01, HBPI-02, HBPI-03 and HBPI-04 were, respectively, 319, 258, 270, 260, 309 and 237 kJ mol$^{-1}$. The obtained values are very close to the ones achieved by the Friedman method. Also, a similar trend is observed in which the $E_a$ decreases as the branching increases. The main advantage of FWO method is that apart from the Arrhenius temperature-dependence, no assumptions regarding the form of the kinetic model equations are required [28].

The third method used was the KAS method in which equation (2.11) relates the temperature ($T$) and the heating rate for a given conversion ($\alpha$) [41,42]. For all the values of $\alpha$, straight line plot of $\ln (\beta/T2)$ against $1/T$ is attained at various $\beta$ [43]. The average activation energy values achieved by the KAS method for LPI-01, LPI-02, HBPI-01, HBPI-02, HBPI-03 and HBPI-04, were 324, 278, 273 264, 315 and 238 kJ mol$^{-1}$, respectively. Again, these values are comparable to the values achieved via the Friedman and FWO techniques. Approximately equal values of $E_a$ were obtained by the methods of FWO and KAS, with slightly higher values obtained by the KAS method. Except for HBPI-03, the activation energy decreases with branching, which shows that the polymer samples with higher branching degraded more easily [24].

Figure 7 presents the relationship between activation energy ($E_a$) and the extent of conversion ($\alpha$) for the Friedman, KAS and FWO approaches. For the linear polymer (LPI-01), there was a decrease in the activation energy with $\alpha$ for all the methods, whereas the branched polymers and LPI-02 are likely to degrade by a multi-step process as indicated by the nonlinear trend of $E_a$ with $\alpha$. It was established that if there is a difference between the $E_a$ attained by the isoconversional methods (integral and differential), then there is a dependency between the activation energy and $\alpha$ [33,66,67]. It was also demonstrated that in both FWO and Friedman methods, different values of $E_a$ are obtained for a system of independent or competitive reaction mechanisms. However, if $E_a$ and $\alpha$ are independent,

then there is a high tendency of obtaining similar values of $E_a$ by these approaches [28,68,69]. The values $E_a$ found via the Friedman method, varied slightly with those obtained via the methods of KAS and the FWO. These dissimilarities are possibly due to the temperature integrals approximations used during the derivation of the equations for the nonlinear isoconversional techniques. The stability order of the polymers studied may be written as LPI-01 > HBPI-03 > HBPI-01 > HBPI-02 > LPI-01 > HBPI-04 based on the values of $E_a$ calculated.

Compared with the average value of $E_a$ (223 kJ mol$^{-1}$) obtained by Chen *et al.* [24], for hyperbranched exopolysaccharide, the values obtained in the present study are slightly higher. Similarly, average $E_a$ value of 90.35 kJ mol$^{-1}$ was reported for anhydride-terminated HBPs with a molecular weight ranging from 16 000 g mol$^{-1}$ to 19 000 g mol$^{-1}$ and molecular weight distribution from 1.21 to 128, respectively [23]. The dissimilarity between the polymers used in the present study and the ones reported in the literature may be explained in terms of the nature of the polymeric chains, molecular weight or the distribution of molecular weights in the samples [45].

The correlation between $E_a$ and $\alpha$ could be estimated via the isoconversional methods without prior assumptions of the model of reaction. Moreover, the isoconversional methods allow for the detection of the multi-step kinetic relationship between the activation energy and the extent of conversion which may not be revealed by other methods such as the Kissinger method [70].

## 3.6. Coats–Redfern method

The Coats–Redfern (CR) technique was chosen to explore the mechanism for the thermal decomposition of these polymers since the activation energy, $E_a$ could be obtained using this approach in the absence of prior information about the nature of the degradation. The activation energy was calculated based on the $f(\alpha)$ functions according to equation (2.12) [44–46].

For each model, the $E_a$ was acquired from the slope of the plot of $\ln(g(\alpha)/T^2)$ versus $1000/T$. Achievement of high correlation coefficient ($R^2$), as well as agreement between the $E_a$ achieved via the CR technique compared with those attained by the isoconversional methods, allows for the selection of the kinetic model [23,46,47]. The $f(\alpha)$ expressions for various mechanisms are given in table 4 and similar $\alpha$-values were adopted as in the isoconversional methods. For both the linear and branched polymers, 12 different $f(\alpha)$ forms were employed to obtain the thermal degradation mechanism (table 5).

Correlation coefficient for the 12 different $f(\alpha)$ were computed and the mechanism of the reaction was selected based on the value of the correlation coefficient, and the right degradation mechanism was assumed to achieve the best linear correlation coefficient, from which the activation energy was determined [71]. However, comparison of the activation energy between the Coats–Redfern method and isoconversional methods for all selected kinetic models could not suggest the accurate mechanism for thermal degradation of these polymers. Nevertheless, analyses of the data together with the high correlation coefficients suggest the most likely reaction mechanisms. The highest correlation for the thermal decomposition of all the polymers (LPI-01, LPI-02, HBPI-01, HBPI-02, HBPI-03 and HBPI-04) was obtained using the function: $f(\alpha) = \alpha^2$, corresponding to a one-dimensional diffusion (D1) mechanism. The activation energies in kJ mol$^{-1}$ for this mechanism were, respectively, 144, 122, 91, 152, 122 and 41 for the LPI-01, LPI-02, HBPI-01, HBPI-02, HBPI-03 and HBPI-04 (table 5).

# 4. Conclusion

Highly branched polymers have been synthesized via polymerization anionic based on the Strathclyde approach under high vacuum conditions. The branched polymers were characterized to be of broad molecular weight by size exclusion chromatography (SEC), whereas the linear polymers prepared in the absence of TMEDA analogue were of narrow molecular weight distribution. DSC and melt rheological measurements explain the nature of branching of the polymers via the polymer segmental motion and chain entanglement phenomena, and it was observed that the branched polymers showed better chain mobility, and a lower complex viscosity than the corresponding linear sample, signifying a lower chain entanglement in the branched polymer molecules. The morphology of the polymers was investigated using HRTEM in order to evaluate the effect of architecture on the morphology of these polymers. It was found that moving from the linear to the highly branched polymer samples, led to the disappearance of the long-range, ordered morphology characteristics of the linear polyisoprene. The high branching nature of the branched polymers is certainly accountable for preventing the long-range lattice order. TGA was employed to study the kinetics of thermal degradation of the linear as well as the highly branched polymers. The TGA experiments were achieved under nitrogen. Three

**Table 5.** Activation energies for the linear and branched polyisoprene obtained by Coats–Redfern.

| Model | LPI-01 $E_a$ (kJ mol$^{-1}$) | $R^2$ | LPI-02 $E_a$ (kJ mol$^{-1}$) | $R^2$ | HBPI-01 $E_a$ (kJ mol$^{-1}$) | $R^2$ | HBPI-02 $E_a$ (kJ mol$^{-1}$) | $R^2$ | HBPI-03 $E_a$ (kJ mol$^{-1}$) | $R^2$ | HBPI-04 $E_a$ (kJ mol$^{-1}$) | $R^2$ |
|---|---|---|---|---|---|---|---|---|---|---|---|---|
| $A_2$ | 43 | 0.980 | 34 | 0.9679 | 26 | 0.9299 | 46 | 0.9809 | 35 | 0.9698 | 06 | 0.8781 |
| $A_3$ | 25 | 0.9740 | 19 | 0.9568 | 13 | 0.8922 | 27 | 0.9759 | 20 | 0.9591 | 01 | 0.8203 |
| $A_4$ | 16 | 0.9658 | 12 | 0.9300 | 07 | 0.8127 | 17 | 0.9685 | 12 | 0.9409 | 02 | 0.911 |
| $D_4$ | 169 | 0.9947 | 140 | 0.9956 | 107 | 0.9765 | 178 | 0.9940 | 143 | 0.9915 | 46 | 0.8846 |
| $D_3$ | 184 | 0.9916 | 152 | 0.9900 | 117 | 0.9691 | 194 | 0.9914 | 155 | 0.9871 | 50 | 0.9662 |
| $D_2$ | 162 | 0.9958 | 135 | 0.9975 | 103 | 0.9800 | 171 | 0.9949 | 137 | 0.9933 | 45 | 0.9697 |
| $D_1$ | 144 | 0.9972 | 122 | 0.9990 | 91 | 0.9872 | 152 | 0.9956 | 122 | 0.9964 | 41 | 0.9756 |
| $F_3$ | 189 | 0.9261 | 152 | 0.9503 | 118 | 0.9709 | 199 | 0.9297 | 158 | 0.9115 | 43 | 0.9121 |
| $F_2$ | 139 | 0.9555 | 112 | 0.9126 | 86 | 0.9081 | 146 | 0.9579 | 116 | 0.9436 | 32 | 0.9534 |
| $F_1$ | 98 | 0.9839 | 79 | 0.9750 | 60 | 0.9504 | 103 | 0.9844 | 82 | 0.9766 | 23 | 0.9576 |
| $R_2$ | 81 | 0.9932 | 71 | 0.9943 | 49 | 0.9695 | 86 | 0.9926 | 67 | 0.9890 | 19 | 0.9287 |
| $R_3$ | 86 | 0.9907 | 70 | 0.9888 | 53 | 0.9635 | 91 | 0.9905 | 72 | 0.9854 | 04 | 0.9468 |

different isoconversional methods, the Friedman, FWO and KAS approaches, were employed to determine the values of $E_a$ for these polymers. The values obtained using the Friedman method varied slightly from the values obtained via the KAS and FWO approaches. The branched polymers were decomposed via multi-step kinetics as manifested by the lack of dependency between the extent of conversion and the activation energy, while for the linear polymer there was a decrease in activation energy with the conversion. The determination of the mechanism via the Coats–Redfern method revealed that the decomposition of polymers can be designated by a D1 mechanism which involves one-dimensional diffusion. This study may offer useful information on the relationship between architecture and polymer properties.

Data accessibility. The data supporting this article has been uploaded at the Dryad Digital Repository: https://doi.org/10.5061/dryad.0cfxpnvx2 [72].

Authors' contribution. S.H. carried out the experiments, analysed the results and prepared the manuscript. N.M.S. was principal investigator, supervised the project and prepared the manuscript. N.A.S. co-supervised the project, analysed the results and contributed to the interpretation of data. M.Z. co-supervised the project, analysed the results and prepared the manuscript. All authors gave final approval for publication.

Competing interests. Authors declare no competing interests.

Funding. This work was supported by the University of Malaya through UM Frontier Research (grant no. FG042-17AFR) and the Ministry of Education through the Fundamental Research Grant Scheme, FRGS (FP136-2019A).

Acknowledgements. The authors gratefully acknowledge the University of Malaya (UM) and the Ministry of Education (MOE) for providing the research facilities and financial support through UM Frontier Research Grant (FG042-17AFR) and the Fundamental Research Grant Scheme, FRGS (FP136-2019A).

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
