## [Reviewer comments · Royal Society Open Science]

Review History

RSOS-190869.R0 (Original submission)

Review form: Reviewer 1

Is the manuscript scientifically sound in its present form?

No

Are the interpretations and conclusions justified by the results?

No

Is the language acceptable?

Yes

Is it clear how to access all supporting data?

Not Applicable

Do you have any ethical concerns with this paper?

No

Have you any concerns about statistical analyses in this paper?

I do not feel qualified to assess the statistics

Recommendation?

Major revision is needed (please make suggestions in comments)

Comments to the Author(s)

The manuscript describes thermal degradation studies of hyperbranched polyisoprene (HBPI). The polymers have been studied by various techniques such as TGA, DSC, DTG and rheological measurements. The thermal analysis data is fitted to several models. The work has been carried out systematically. Average activation energy for degradation was calculated from different kinetic methods. However, clear insight into the mechanism for degradation is not obtained by comparison of activation energy between different methods for all kinetic models. An understanding of the effect of branching on decomposition is not developed due to lack of series of polymers with controlled molecular parameters.

Research on HB polymers is of interest to academia as well as industry due to their practical applications. The results in this manuscript would be of interest to researchers working in the area of thermal analysis of polymers. However, certain concerns need to be addressed before the manuscript could be recommended for publication in Royal Society Open Science. Specific comments are given below.

1. Could the authors calculate the degree of branching (DB) for the HB polymers? It appears that the linear component in HBPI-02 is large. The dispersity is also very high. This may be why the activation energy for linear polymer and HBPI-02 is nearly equal.
2. Related to point no. 1, the viscosity for polymer with supposedly higher DB (HBPI-02) is higher than that for polymer with lower DB (HBPI-01) (Table 1).
3. More polymers with controlled branching need to be synthesized and thoroughly characterized for molecular architecture. At least one more HB polymer should be made to replace HBPI-02.
4. There is no unit for activation energy in Table 1.

Review form: Reviewer 2**Is the manuscript scientifically sound in its present form?**

No

Are the interpretations and conclusions justified by the results?

No

Is the language acceptable?

Yes

Do you have any ethical concerns with this paper?

No

Recommendation?

Major revision is needed (please make suggestions in comments)

Comments to the Author(s)

Sarih and coworkers reported the thermal degradation properties of hyperbranched polyisoprenes (PIs) via model analysis on their TGA data. The rheological and thermal degradation properties of hyperbranched polymers (HBPs) are important for their practical application, as well as for scientific interests. The topic seems interesting, but the manuscript should be improved to be published, possibly with a series of samples (polymers). The most critical part is that it seems hard to draw a conclusion with only three samples: linear, less branched, and highly branched (and high dispersity). As the authors mentioned, the rheological and thermal properties can be affected by branching density/index and polydispersity. However, the third sample is both highly branched AND of broad dispersity at the same time, confusing to find out which factor is working. In their previous report (ref 1), they already synthesized 6 linear and 10-12 branched PIs with different branching and dispersity indexes. It is wondering why they selected those three samples only, and if a few more samples were added, more clear conclusions could be drawn and the paper becomes complete. Thus for publication, the manuscript should be supplemented by more data, and the following issues should be addressed as well.

1. As stated above, the authors already published the synthesis and characterization of highly branched polyisoprenes, comparing those of linear ones. Thus Figure 1 (GPC), Table 1 (molecular characteristics), Figure 2 (1H-NMR), and Figure 4-5 (rheological characteristics) seem repeated since the molecular characteristics as shown in Table 1 are exactly same. Is there any particular reason to report those things again? They should be summarized succinctly or provide with specific reasons if any.

In addition, the presentations regarding the molecular structures are insufficient for readers to follow (for example, peaks a-c in 1H-NMR). The molecular structures or characteristics appear very important in the discussion, and thus they should be clearly presented.

2. In the analysis of TEM images (Figure 3), the authors drew some conclusions regarding long-range ordering but the quality of each image seems different. Since TEM images are very "local", the most representative image with the similar quality should be compared. Furthermore, for long-range ordering, it would be more helpful if some quantitative data are provided.

3. In TGA (Figure 6), the weight loss at around 200-250 °C of HBPI-01 is quite different from that of other samples, which is not easy to understand. However, in Figure 7, that difference is not evident and rather HBPI-02 showed more decrease. What is the reason? Can more reproducible data be provided?

4. The authors used three different methods (Friedman, FWO, and KAS) to obtain average E_a . For the readers sake, it would be helpful if the authors can introduce those methods in a brief way (What are the differences? What are the strength and weakness?)

5. The mechanism of thermal decomposition of PIs were analyzed using the Coats-Redfern technique, but the descriptions seem insufficient. What are "f(a) expressions for various mechanisms"? Rather than just mathematical expression, the molecular descriptions should be provided. Furthermore, more discussion regarding the one-dimensional diffusion mechanism should be supplemented.

6. Experimental section contains many typo, which the authors should check out thoroughly.

Decision letter (RSOS-190869.R0)

25-Jun-2019

Dear Dr Habibu:

Title: Rheological and thermal degradation properties of hyperbranched polyisoprene prepared by anionic polymerisation

Manuscript ID: RSOS-190869

The editor assigned to your manuscript has now received comments from reviewers. We would like you to revise your paper in accordance with the referee and Subject Editor suggestions which can be found below (not including confidential reports to the Editor). Please note this decision does not guarantee eventual acceptance.

Please submit your revised paper before 18-Jul-2019. Please note that the revision deadline will expire at 00.00am on this date. If we do not hear from you within this time then it will be assumed that the paper has been withdrawn. In exceptional circumstances, extensions may be possible if agreed with the Editorial Office in advance. We do not allow multiple rounds of revision so we urge you to make every effort to fully address all of the comments at this stage. If deemed necessary by the Editors, your manuscript will be sent back to one or more of the original reviewers for assessment. If the original reviewers are not available we may invite new reviewers.

Please also include the following statements alongside the other end statements. As we cannot publish your manuscript without these end statements included, if you feel that a given heading is not relevant to your paper, please nevertheless include the heading and explicitly state that it is not relevant to your work.

- Funding statement

Please include a funding section after your main text which lists the source of funding for each author.

RSC Associate Editor:
Comments to the Author:
(There are no comments.)

RSC Subject Editor:
Comments to the Author:
(There are no comments.)

Reviewers' Comments to Author:
Reviewer: 1

Comments to the Author(s)

The manuscript describes thermal degradation studies of hyperbranched polyisoprene (HBPI). The polymers have been studied by various techniques such as TGA, DSC, DTG and rheological measurements. The thermal analysis data is fitted to several models. The work has been carried out systematically. Average activation energy for degradation was calculated from different kinetic methods. However, clear insight into the mechanism for degradation is not obtained by comparison of activation energy between different methods for all kinetic models. An understanding of the effect of branching on decomposition is not developed due to lack of series of polymers with controlled molecular parameters.

Research on HB polymers is of interest to academia as well as industry due to their practical applications. The results in this manuscript would be of interest to researchers working in the area of thermal analysis of polymers. However, certain concerns need to be addressed before the manuscript could be recommended for publication in Royal Society Open Science. Specific comments are given below.

1. Could the authors calculate the degree of branching (DB) for the HB polymers? It appears that the linear component in HBPI-02 is large. The dispersity is also very high. This may be why the activation energy for linear polymer and HBPI-02 is nearly equal.
2. Related to point no. 1, the viscosity for polymer with supposedly higher DB (HBPI-02) is higher than that for polymer with lower DB (HBPI-01) (Table 1).
3. More polymers with controlled branching need to be synthesized and thoroughly characterized for molecular architecture. At least one more HB polymer should be made to replace HBPI-02.
4. There is no unit for activation energy in Table 1.

Reviewer: 2

Comments to the Author(s)

Sarih and coworkers reported the thermal degradation properties of hyperbranched polyisoprenes (PIs) via model analysis on their TGA data. The rheological and thermal degradation properties of hyperbranched polymers (HBPs) are important for their practical application, as well as for scientific interests. The topic seems interesting, but the manuscript should be improved to be published, possibly with a series of samples (polymers). The most critical part is that it seems hard to draw a conclusion with only three samples: linear, less

branched, and highly branched (and high dispersity). As the authors mentioned, the rheological and thermal properties can be affected by branching density/index and polydispersity. However, the third sample is both highly branched AND of broad dispersity at the same time, confusing to find out which factor is working. In their previous report (ref 1), they already synthesized 6 linear and 10-12 branched PIs with different branching and dispersity indexes. It is wondering why they selected those three samples only, and if a few more samples were added, more clear conclusions could be drawn and the paper becomes complete. Thus for publication, the manuscript should be supplemented by more data, and the following issues should be addressed as well.

1. As stated above, the authors already published the synthesis and characterization of highly branched polyisoprenes, comparing those of linear ones. Thus Figure 1 (GPC), Table 1 (molecular characteristics), Figure 2 (1H-NMR), and Figure 4-5 (rheological characteristics) seem repeated since the molecular characteristics as shown in Table 1 are exactly same. Is there any particular reason to report those things again? They should be summarized succinctly or provide with specific reasons if any.

In addition, the presentations regarding the molecular structures are insufficient for readers to follow (for example, peaks a-c in 1H-NMR). The molecular structures or characteristics appear very important in the discussion, and thus they should be clearly presented.

2. In the analysis of TEM images (Figure 3), the authors drew some conclusions regarding long-range ordering but the quality of each image seems different. Since TEM images are very "local", the most representative image with the similar quality should be compared. Furthermore, for long-range ordering, it would be more helpful if some quantitative data are provided.

3. In TGA (Figure 6), the weight loss at around 200-250 oC of HBPI-01 is quite different from that of other samples, which is not easy to understand. However, in Figure 7, that difference is not evident and rather HBPI-02 showed more decrease. What is the reason? Can more reproducible data be provided?

4. The authors used three different methods (Friedman, FWO, and KAS) to obtain average E_a . For the readers sake, it would be helpful if the authors can introduce those methods in a brief way (What are the differences? What are the strength and weakness?)

5. The mechanism of thermal decomposition of PIs were analyzed using the Coats-Redfern technique, but the descriptions seem insufficient. What are "f(a) expressions for various mechanisms"? Rather than just mathematical expression, the molecular descriptions should be provided. Furthermore, more discussion regarding the one-dimensional diffusion mechanism should be supplemented.

6. Experimental section contains many typo, which the authors should check out thoroughly.

Author's Response to Decision Letter for (RSOS-190869.R0)

See Appendix A.

RSOS-190869.R1 (Revision)

Review form: Reviewer 2

Is the manuscript scientifically sound in its present form?

No

Are the interpretations and conclusions justified by the results?

No

Is the language acceptable?

Yes

Do you have any ethical concerns with this paper?

No

Have you any concerns about statistical analyses in this paper?

No

Recommendation?

Major revision is needed (please make suggestions in comments)

Comments to the Author(s)

The authors did not respond to the most critical issue raised by this reviewer – “The topic seems interesting, but the manuscript should be improved to be published, possibly with a series of samples (polymers). The most critical part is that it seems hard to draw a conclusion with only three samples: linear, less branched, and highly branched (and high dispersity). As the authors mentioned, the rheological and thermal properties can be affected by branching density/index and polydispersity. However, the third sample is both highly branched AND of broad dispersity at the same time, confusing to find out which factor is working. In their previous report (ref 1), they already synthesized 6 linear and 10-12 branched PIs with different branching and dispersity indexes. It is wondering why they selected those three samples only, and if a few more samples were added, more clear conclusions could be drawn and the paper becomes complete.”

In short, 3 or 4 samples seem insufficient to draw conclusion since they already have more than 16 samples, which would make this work more complete. The authors should address this issue. In addition, the corrections and amendments are not highlighted and unable to be distinguished. The authors need to upload the highlighted version too.

Decision letter (RSOS-190869.R1)

30-Jul-2019

Dear Dr Habibu:

Title: Rheological and thermal degradation properties of hyperbranched polyisoprene prepared by anionic polymerisation

Manuscript ID: RSOS-190869.R1

The editor assigned to your paper has now received comments from reviewers. We would like you to revise your paper in accordance with the referee and Subject Editor suggestions which can be found below (not including confidential reports to the Editor). Please note this decision does not guarantee eventual acceptance.

Please submit a copy of your revised paper before 22-Aug-2019. Please note that the revision deadline will expire at 00.00am on this date. If we do not hear from you within this time then it will be assumed that the paper has been withdrawn. In exceptional circumstances, extensions may be possible if agreed with the Editorial Office in advance. We do not allow multiple rounds of revision so we urge you to make every effort to fully address all of the comments at this stage. If deemed necessary by the Editors, your manuscript will be sent back to one or more of the original reviewers for assessment. If the original reviewers are not available we may invite new reviewers.

RSC Associate Editor:
Comments to the Author:
(There are no comments.)

RSC Subject Editor:
Comments to the Author:
(There are no comments.)

Reviewers' Comments to Author:

Reviewer: 2

Comments to the Author(s)

The authors did not respond to the most critical issue raised by this reviewer - "The topic seems interesting, but the manuscript should be improved to be published, possibly with a series of samples (polymers). The most critical part is that it seems hard to draw a conclusion with only three samples: linear, less branched, and highly branched (and high dispersity). As the authors mentioned, the rheological and thermal properties can be affected by branching density/index and polydispersity. However, the third sample is both highly branched AND of broad dispersity at the same time, confusing to find out which factor is working. In their previous report (ref 1), they already synthesized 6 linear and 10-12 branched PIs with different branching and dispersity indexes. It is wondering why they selected those three samples only, and if a few more samples were added, more clear conclusions could be drawn and the paper becomes complete."

In short, 3 or 4 samples seem insufficient to draw conclusion since they already have more than 16 samples, which would make this work more complete. The authors should address this issue. In addition, the corrections and amendments are not highlighted and unable to be distinguished. The authors need to upload the highlighted version too.

Author's Response to Decision Letter for (RSOS-190869.R1)

See Appendix B.

RSOS-190869.R2 (Revision)

Review form: Reviewer 2

Is the manuscript scientifically sound in its present form?

Yes

Are the interpretations and conclusions justified by the results?

Yes

Is the language acceptable?

Yes

Do you have any ethical concerns with this paper?

No

Have you any concerns about statistical analyses in this paper?

No

Recommendation?

Accept as is

Comments to the Author(s)

The issues seem to be reasonably well addressed by the authors, which is now acceptable.

Decision letter (RSOS-190869.R2)

04-Oct-2019

Dear Dr Habibu:

Title: Rheological and thermal degradation properties of hyperbranched polyisoprene prepared by anionic polymerisation

Manuscript ID: RSOS-190869.R2

It is a pleasure to accept your manuscript in its current form for publication in Royal Society Open Science. The chemistry content of Royal Society Open Science is published in collaboration with the Royal Society of Chemistry.

RSC Associate Editor:
Comments to the Author:
(There are no comments.)

RSC Subject Editor:
Comments to the Author:
(There are no comments.)

Reviewer(s)' Comments to Author:
Reviewer: 2

Comments to the Author(s)
The issues seem to be reasonably well addressed by the authors, which is now acceptable.

Appendix A

RESPONSE TO REVIEWERS COMMENTS

First, we would like to express our gratitude to the reviewers and Journal Editor for sparing their precious time to review our manuscript rigorously and for making valuable comments. We have made all the necessary corrections as given in our response in the table below. For easy identification and tracking, changes and modifications made in the manuscript were highlighted blue fonts for addition and red/strike through fonts for deletion.

No.	Reviewers Comments	Authors' Response
Reviewer #1		
1.	Could the authors calculate the degree of branching (DB) for the HB polymers? It appears that the linear component in HBPI-02 is large. The dispersity is also very high. This may be why the activation energy for linear polymer and HBPI-02 is nearly equal.	The branching factor (g') is a measure of the degree of branching. The lower the g' value, the higher the degree of branching and vice-versa (See Table 1). Moreover, high dispersity is one of the characteristics of hyperbranched polymers, and of course the high dispersity index may lower the activation energy due to the presence of the low molecular weight component in the hyperbranched polymer sample.
2.	Related to point no. 1, the viscosity for polymer with supposedly higher DB (HBPI-02) is higher than that for polymer with lower DB (HBPI-01) (Table 1).	The complex viscosity for HBPI-02 is lower than that of HBPI-01 (Figure 4), and this expected since the HBPI-02 has lower g' (0.37) which implies higher DB.
3.	More polymers with controlled branching need to be synthesized and thoroughly characterized for molecular architecture. At least one more HB polymer should be made to replace HBPI-02.	We have included one more HB polymer sample (HBPI-03) as suggested.
4.	There is no unit for activation energy in Table 1.	The unit (kJmol^{-1}) has been added to the Table. However, it is supposed to be Table 3 instead of Table 1 (now Table 4). The error has been corrected
Reviewer #2		
1.	As stated above, the authors already published the synthesis and characterization of highly branched	We have removed Figure 1 since it appears that same information regarding molecular characteristics is repeated in Table 1. For the rheological characteristics, more samples are considered in the

	polyisoprenes, comparing those of linear ones. Thus Figure 1 (GPC), Table 1 (molecular characteristics), Figure 2 (¹H-NMR), and Figure 4-5 (rheological characteristics) seem repeated since the molecular characteristics as shown in Table 1 are exactly same. Is there any particular reason to report those things again? They should be summarized succinctly or provide with specific reasons if any. In addition, the presentations regarding the molecular structures are insufficient for readers to follow (for example, peaks a-c in ¹H-NMR). The molecular structures or characteristics appear very important in the discussion, and thus they should be clearly presented.	present manuscript compared to the previous one. More discussion has been given on the ¹H-NMR and the molecular structure.
2.	In the analysis of TEM images (Figure 3), the authors drew some conclusions regarding long-range ordering, but the quality of each image seems different. Since TEM images are very “local”, the most representative image with the similar quality should be compared. Furthermore, for long-range ordering, it would be more helpful if some quantitative data are provided.	TEM image for HBPI-01 has been removed from Figure 3 to allow comparison of the linear and HBPI-02 as the most representative samples with similar and better quality. We were not able to conduct quantitative analysis for the long-range ordering.
3.	In TGA (Figure 6), the weight loss at around 200-250 °C of HBPI-01 is quite different from that of other samples, which is not easy to understand. However, in Figure 7, that difference is not evident and rather HBPI-02 showed more decrease. What is the reason? Can more reproducible data be provided.	Figure 6 presented the weight loss at 15 °C min⁻¹ of different samples but in Figure 7, various heating rates are presented as such the weight loss at around 200-250 °C might not be as visible as the presented in Figure 6 possibly due to some overlap with other heating rates.
4.	The authors used three different methods (Friedman, FWO, and KAS) to obtain average E_a. For the readers sake, it would be helpful if the authors can introduce those methods in a brief way (What are the differences? What are the strength and weakness?)	Brief introduction of the kinetic methods (Friedman, FWO, and KAS) has now been provided. The relevant section of the manuscript has been updated.

5.	The mechanism of thermal decomposition of PIs were analyzed using the Coats-Redfern technique, but the descriptions seem insufficient. What are “f(a) expressions for various mechanisms”? Rather than just mathematical expression, the molecular descriptions should be provided. Furthermore, more discussion regarding the one dimensional diffusion mechanism should be supplemented.	Explanation on various f(α) expressions has been given. Table 1 has been added and relevant portion of Table 4 has been modified. The numbering of Tables throughout the manuscript has been adjusted. Further discussion regarding the one dimensional diffusion mechanism has also been given.
6.	Experimental section contains many typos, which the authors should check out thoroughly.	We have checked and corrected the typos in the experimental section accordingly.
Thank you for all the comments		

Appendix B

Department of Chemistry,
Faculty of Science,
University of Malaya
50603, Kuala Lumpur,
Malaysia
19th September 2019

Professor Jeremy Sanders,
Editor-in-chief,
Royal Society Open Science,

Dear Professor Jeremy,

Re: Manuscript (RSOS-190869-R2)

Please find attached the revised version of our manuscript ***Rheological and thermal degradation properties of hyperbranched polyisoprene prepared by anionic polymerisation*** which we would like to resubmit for publication as a research article in the Royal Society Open Science journal.

The reviewers' comments are highly insightful and enabled us to improve the quality of our manuscript greatly. Enclosed are our responses to the reviewer's comments. Revisions in the text are highlighted in yellow.

Hope our responses would be sufficient to make our manuscript suitable for publication.

Yours sincerely,

Shehu Habibu